# The Role of Chitinases in Chronic Airway Inflammation Associated with Tobacco Smoke Exposure

**DOI:** 10.3390/cells11233765

**Published:** 2022-11-25

**Authors:** Natalia Przysucha, Katarzyna Górska, Marta Maskey-Warzęchowska, Małgorzata Proboszcz, Patrycja Nejman-Gryz, Magdalena Paplińska-Goryca, Barbara Dymek, Agnieszka Zagozdzon, Rafał Krenke

**Affiliations:** 1Department of Internal Medicine, Pulmonary Diseases and Allergy, Medical University of Warsaw, 02-097 Warsaw, Poland; 2Postgraduate School of Molecular Medicine, Medical University of Warsaw, 02-097 Warsaw, Poland; 3Molecure SA, 02-089 Warsaw, Poland

**Keywords:** COPD, airway inflammation, chitinase, cytokines, YKL-40, CHIT1

## Abstract

Chitinases and chitinase-like proteins are thought to play a role in innate inflammatory responses. Our study aimed to assess whether chitinase concentration and activity in induced sputum (IS) of patients exposed to tobacco smoke are related to the level of airway inflammation including the level and activity of chitinases and chitinase-like proteins. The study included 22 patients with chronic obstructive pulmonary disease (COPD), 12 non-COPD smokers, and nine nonsmoking subjects. Sputum CHIT1 and YKL-40 levels and chitinolytic activity were compared with sputum IL-6, IL-8, IL-18, and MMP-9 levels. A hierarchical cluster analysis was also performed. Sputum YKL-40 was higher in COPD patients than in the control groups. Sputum CHIT1 and YKL-40 levels correlated with IS inflammatory cell count as well as with MMP-9 and IL-8 levels. Two main clusters were revealed: Cluster 1 had lower chitinase levels and activity, lower IS macrophage and neutrophil count, and lower IS IL-8, IL-18, and MMP-9 than Cluster 2. Comparison of COPD patients from both clusters revealed significant differences in the IS inflammatory profile despite comparable clinical and functional data. Our findings seem to confirm the involvement of chitinases in smoking-associated chronic airway inflammation and show that airway chitinases may be a potential novel marker in COPD phenotyping.

## 1. Introduction

Chitinases, represented by chitotriosidase (CHIT1) and acid mammalian chitinase (AMCase), are a group of proteins involved in the innate immune response to chitin-containing organisms (i.e., parasites and fungi) [1,2]. They function by binding to chitin chains and hydrolyzing them to low molecular weight chitooligomers [3]. Chitinase-like proteins (CLPs) are a group of proteins which do not have hydrolytic activity but are still capable of binding chitin or chitin particles [3]. The most prominent CLP representant is chitinase-3-like protein 1 (CHI3L1), also known as YKL-40 [1]. An imbalance of chitinase activity has been reported in several inflammatory lung disorders, including asthma, chronic obstructive pulmonary disease (COPD), and fibrosing lung diseases [1,2,3,4,5]. Recent studies are focused not only on the role of chitinases and CLPs in airway diseases but also on chitinases as a therapeutic target [6,7].

The primary cell sources of CHIT1 and YKL-40 in the lung are macrophages and neutrophils [8,9]. YKL-40 is also produced by epithelial cells [9]. Multiple triggers, including environmental toxins and tobacco smoke exposure, may enhance chitinase and CLP production [1]. It has been shown that CHIT1 and YKL-40 may simultaneously stimulate alveolar macrophages and epithelial cells to secrete proinflammatory and profibrotic mediators resulting in tissue inflammation, alveolar destruction, and tissue remodeling [1,5]. Smoking is known to be associated with an upregulation of chitinase gene expression and chitinase levels were found to be higher in smokers than those who had never smoked. This phenomenon was demonstrated not only in studies comparing patients with airway diseases with both smokers and non-smokers [8,10], but also in studies conducted in smoking and non-smoking patients affected by extrapulmonary disorders [11,12].

The role of chitinases in the pathogenesis of COPD is a relatively novel point of interest. With the growing evidence for their contribution to COPD-related inflammatory pathways and their potential application as a marker for smokers at risk of developing the disease [12,13], chitinases are a promising area of research. However, there is still little known about the mutual relationships between chitinases and other COPD inflammatory markers. Another understudied area is the role of chitinases and CLPs as potential novel treatment targets for COPD; considering that the efficacy of chitinase inhibitors is currently being investigated for the treatment of lung diseases in in vitro and animal models [14] and that therapeutic options in COPD are still limited and unsatisfactory in terms of effectiveness, targeting chitinases and CLPs is an attractive possible solution.

In the light of the above, we undertook a study aimed at assessing the levels and activity of CHIT1 and YKL-40 in induced sputum, as well as the mutual relationships between CHIT1/YKL-40 and various features of airway inflammation in patients with COPD, smokers without COPD, and people who have never smoked. Our working hypothesis, based on previously published data and our own results, was that chitinases/CLPs are not specifically related to COPD, but rather to the level of airway inflammation due to exposure to noxious factors, mainly those present in tobacco smoke. Thus, the specific aims were to: (1) compare chitinase concentration and activity in induced sputum (IS) in the investigated groups, (2) search for potential relationships between sputum chitinase concentration/activity and other inflammatory biomarkers, and (3) correlate chitinase concentration/activity with the clinical profiles of the investigated patients.

## 2. Materials and Methods

### 2.1. General Study Design and Participants

This prospective, observational study was performed by the Department of Internal Medicine, Pulmonary Diseases and Allergy of the Medical University of Warsaw between May 2018 and February 2019 and included mild-to-moderate COPD patients and controls who agreed to participate in the project. The ethics committee of the Medical University of Warsaw (KB/67/A/2018) approved the study. All participants signed an informed consent form before participating.

The scope of this study included: medical history, with particular attention given to signs and symptoms assessed using the modified Medical Research Council (mMRC) scale and the COPD Assessment Test (CAT), smoking and exacerbations history, physical examination, spirometry, and sputum induction.

### 2.2. Subjects

Forty-three subjects were enrolled, including 22 patients with COPD, 12 smokers without COPD, and 9 healthy nonsmokers. Inclusion criteria were as follows: (1) age above 40 years, and (2) diagnosis of COPD based on past medical history, data on noxious inhalants exposure, typical signs and symptoms, and irreversible airway obstruction identified by spirometry. We defined airway obstruction using a z-score of −1.645 (5th percentile) as the lower limit of normal [15] and divided COPD patients into four categories (A–D) depending on symptom severity and exacerbation history as recommended in the Global Initiative for Chronic Obstructive Lung Disease (GOLD) 2018 guidelines [16].

The inclusion criteria applied for the control groups (smokers and nonsmokers) comprised of: age above 40 years, a negative history of any chronic lung diseases, and an absence of respiratory symptoms and normal spirometry. Smokers were defined as individuals with 10 or more pack-years of smoking history. Patients who had previously smoked 10 or more pack-years but denied smoking one year before the study onset were defined as ex-smokers. Patients with no smoking history (less than 1 pack-year) were classified as non-smokers. 

Universal exclusion criteria for all subgroups included: symptoms of acute respiratory tract infection in the previous 6 weeks, oral and inhaled steroid treatment within 6 weeks preceding sample collection, active cancer disease, and any worsening of comorbidities (i.e., heart failure) within 6 weeks before the examination. 

### 2.3. Spirometry

Spirometry testing was performed using a Lungtest 1000 spirometer (MES, Cracow, Poland) according to the American Thoracic Society/European Respiratory Society (ATS/ERS) guidelines [17]. Pre- and post-bronchodilator forced expiratory volume in the first second (FEV_1_), forced vital capacity (FVC), and FEV_1_/FVC% were measured in all COPD patients, while only baseline measurements were performed in control subjects (no bronchodilators administered). 

### 2.4. Induced Sputum Collection and Processing

Sputum induction was preceded by inhalation of 400 μg of salbutamol. Next, inhalations of sterile hypertonic saline (NaCl) were applied at increasing concentrations (3, 4, and 5% solutions) via an ultrasonic nebulizer (ULTRA-NEBTM2000, DeVilbiss, Somerset, PA, USA), following ERS recommendations [18]. Induced sputum samples were processed as described elsewhere [19]. Sputum smears were used to assess total and differential leukocyte count. Induced sputum supernatants were stored at −80 °C and then used to measure cytokine concentrations, YKL-40 and CHIT1 levels, and chitinolytic activity. 

### 2.5. Cytokine and Chitinase Concentration Measurements in IS

IL-6, IL-8, IL-18, MMP-9, YKL-40, and CHIT1 concentrations were measured using a standard quantitative sandwich enzyme linked immunosorbent assay (ELISA) technique, strictly according to the manufacturer’s instructions (R&D Systems, Minneapolis, MN, USA, and CircuLex Human ELISA Kit, MBL, Woburn, MA, USA). Each sample was evaluated twice. The IL-6 assay range was 3.1–300 pg/mL and the sensitivity was 0.7 pg/mL. For the IL-8, IL-18, and MMP-9 assays, the respective values for the assay range and sensitivity were as follows: 31.2–2.0 pg/mL and 7.5 pg/mL, 15.6–1.0 pg/mL and 5.15 pg/mL, and 0.3–20 ng/mL and 0.156 ng/mL. The human CHIT1 and YKL-40 detection ranges were 56.25–3600 pg/mL and 62.5–4000 pg/mL, respectively, and the sensitivity values were 48.3 pg/mL and 8.15 pg/mL, respectively.

### 2.6. Chitinolytic Activity Analysis

Chitinolytic activity was detected using the procedure described previously [20]. The concentrations of some substrates and volumes of analyzed samples were modified to adjust the procedure to human materials [6]. Substrate hydrolysis product 4-methlyumberlliferone was measured fluorometrically using a Tecan Spark 10M microplate reader (Tecan, Switzerland) (excitation 355 nm/emission 460 nm). Chitinolytic activity in human samples was measured at pH 2 for the AMCase activity only and at pH 6 for both AMCase and CHIT1. Each sample was tested twice.

### 2.7. Statistical Analysis

Estimation of the sample size was based on data from an earlier study by Létuvé S et al. [21]. To detect between-group difference with a power of 80% and a significance level of 5%, the required sample size was estimated to be 42 subjects (21 COPD patients and 21 controls).

Statistical analysis was performed using Statistica 13.3 software (StatSoft Inc., Tulsa, OK, USA) or R environment (version 4.0.5; R Foundation for Statistical Computing, Vienna, Austria). Data were presented as median and interquartile range (IQR) or percentage (%). Differences between groups were tested using a Pearson’s chi-squared test for categorical variables, a Mann–Whitney U test for comparisons between two independent groups, or a Kruskal–Wallis test followed by a Dunn’s post hoc test for multiple comparisons between groups. A *p*-value of less than 0.05 was accepted as indicating significance. Correlations between data were assessed using Spearman rank tests. Hierarchical clustering was performed on standardized data using Euclidean distance Ward D linkage. Outliers were normalized to mean +/−3SD (standard deviations). Logistic regression was performed using data prepared using the same method as for hierarchical clustering (i.e., standardized and outliers clipped). 

## 3. Results

Basic comparative characteristics of the study participants are shown in Table 1.

Data on cellular compositionconcentrations of cytokines, chitinases, and chitinolytic activity in IS are given in Table 2. Patients with COPD were characterized by the highest sputum IL-8 and YKL-40 levels (Table 2 and Figure 1). YKL-40 was undetectable in 75% of smoking controls, 33% of non-smoking controls, and only 14% of COPD patients. There were no differences in CHIT1 concentration and chitinolytic activity between the investigated groups (Table 2 and Figure 1). Chitinolytic activity was detected only at pH 6 and was highly correlated with sputum CHIT1 levels (r = 0.71, *p* < 0.05), suggesting that CHIT1 is the primary source of chitinolytic activity in the airways.

Using a logistic regression model with selected clinical and biochemical parameters that we created, we found that YKL-40 is an important inflammatory factor in COPD pathobiology (r-coefficient of 1.69 and *p*-value of 0.02) (Table 3). 

Analysis of the entire investigated group revealed a significant negative correlation between IS YKL-40 and spirometry parameters (FEV_1_% of predicted and FEV_1_/FVC %, r = −0.56, r = −0.66, respectively, both *p* < 0.0001). Chitinase levels and activity in IS did not correlate with any other clinical features (Figure 2).

A significant correlation between sputum CHIT1, YKL-40, and chitinolytic activity and both total cell count and the number of sputum neutrophils was found. Moreover, significant positive correlations were found between sputum macrophage counts, CHIT1 level, and chitinolytic activity. There were also positive correlations between IS MMP-9 and chitinase levels and chitinolytic activity, and between sputum IL-8 and YKL-40 levels (Figure 2).

A clustering analysis involving IS data from COPD patients and control groups was performed to discover any differences in the features of airway inflammation (Figure 3). Two main clusters were revealed; the clinical characteristics of both clusters are shown in Table 4.

Cluster 1 included patients with COPD (38%), control smokers (35%), and non-smokers (27%) with a preserved gender balance. Cluster 2 included only patients with COPD (100%) and showed male predominance. There was no difference in age and BMI between the two clusters. Patients from Cluster 2 had worse lung function and greater tobacco smoke exposure (Table 4). This cluster also had a significantly higher proportion of total sputum cells and higher macrophage and neutrophil counts. Moreover, chitinase concentration (CHIT1 and YKL-40) and chitinase activity were significantly higher in Cluster 2 than in Cluster 1. The concentration of sputum cytokines IL-8, IL-18, and MMP-9 was elevated in Cluster 2, but the level of IL-6 was higher in Cluster 1 (Table 5).

In the following step, we compared the clinical and biochemical profiles of patients with COPD from Clusters 1 and 2. No differences in sex distribution, age, BMI, smoking exposure, or exacerbation frequency and symptom severity (as reflected by mMRC and CAT scores) were found. There was no statistically significant difference in spirometry test results (Table 6).

Patients with COPD from Cluster 2 were characterized by a significantly higher total IS cell count and a greater number of sputum neutrophils and macrophages. They also had higher levels of sputum IL-8, IL-18, and MMP-9 and higher chitinase levels and activity. Interestingly, the concentration of CHIT1 in sputum COPD patients from Cluster 1 was below the detection threshold (Table 7).

## 4. Discussion

Our study showed an interesting relationship between the level of airway inflammation due to chronic tobacco smoke exposure, expressed as the number of different inflammatory cells in IS, and both CHIT1 and YKL-40 levels and chitinolytic activity. This was demonstrated by a significant positive correlation between the sputum total cell, neutrophil, and macrophage counts and the levels of sputum YKL-40, CHIT-1, and chitinolytic activity in all the investigated groups. Moreover, our study revealed a significant positive correlation between the concentration of sputum MMP-9 and IL-8 and the levels of sputum CHIT1, YKL-40 levels, and CHIT1 activity. Sputum YKL-40 was significantly higher in patients with COPD than in both smoking and non-smoking controls, while CHIT1 and chitinolytic activity levels were comparable in the three investigated groups. A logistic regression model including a combination of clinical and biochemical variables confirmed that elevated IS YKL-40 level is a risk factor associated with COPD.

At the same time, cluster analysis showed that significant differences in chitinase concentration and activity among patients with COPD may occur. To our knowledge, hierarchical clustering analysis was not used in previous studies on chitinases in lung diseases and our study is the first to apply this approach. By demonstrating significant differences between the identified clusters, we confirmed different inflammatory patterns in the disease, supporting a potential role of precision medicine in COPD management.

Bearing in mind that macrophages, neutrophils, and epithelial cells are the main source of airway chitinases [8,9], the positive correlation between sputum cell count, neutrophils, and macrophages and chitinase levels and activity could be expected, however the existing data for COPD are not unequivocal. Otsuka et al. showed that sputum YKL-40 correlated with sputum neutrophils in both COPD and asthma and with sputum macrophages in COPD, although such an association was not found for controls [9]. In a study comparing sputum biomarkers in idiopathic pulmonary fibrosis (IPF) in which patients with COPD comprised one of the control groups, Guiot et al. demonstrated that sputum neutrophil count was significantly higher in patients with IPF and those with COPD compared to healthy controls, however, both sputum levels of YKL-40 and YKL-40 expression did not differ between the groups [22]. Majewski et al., who compared patients with COPD with smoking and non-smoking controls, found a higher sputum neutrophil and macrophage count in patients with COPD vs. non-smoking controls. This was also the case with sputum YKL-40 and both chitotriosidase levels and chitotriosidase expression, but not with the expression of YKL-40 [10]. It should be emphasized that our study is different than all previous studies on airway chitinase/CLPs levels in terms of the general approach. Namely, we were not focused on the chitinase and CLP levels in specific airway diseases but rather on their levels and activities in patients exposed to noxious gases and fumes that can induce and sustain airway inflammation. This was a consequence of our working hypothesis that the level of chitinases/CLPs in the airways is less associated with the specific disease, i.e., COPD, and more with the level of airway inflammation. As we found that participants with chronic airway inflammation (manifested by a higher number of total sputum cells, macrophages, and neutrophils) had significantly higher chitinase concentrations and activity, we believe our results can be construed as supporting this hypothesis. This observation also seems to align with the results of some earlier studies which suggest that CHIT1 and YKL-40 can be markers of inflammation associated with fibrosis and matrix remodeling in the lungs [8,22]. The lack of correlation between the level of cigarette smoking exposure and chitinase levels probably results from the complexity and diversity of airway inflammation. Tobacco smoke exposure is one of the numerous triggers and driving factors of airway inflammation. Moreover, the inflammatory effect of noxious gases and inhaled particles varies and depends on the individual susceptibility and characteristics of airway inflammatory responses.

Our analysis of chitotriosidase and YKL-40 in the context of other sputum inflammatory markers showed a significant positive correlation between CHIT1, YKL40, and chitinolytic activity and MMP-9 and a positive correlation between YKL-40 and IL-8. Furthermore, one of the two identified clusters characterized by elevated chitinase levels and chitinolytic activity also had significantly higher sputum MMP-9, IL-8, and IL-18 levels. These results are in line with earlier findings on the relationship between YKL-40, MMP-9, and IL-8. It has been documented that bronchial epithelial cells treated with YKL-40 show increased IL-8 production [23] and exposure of alveolar macrophages to YKL-40 or CHIT1 promoted the release of IL-8 and MMP-9 [8,22]. Our results confirm these observations.

Cluster analysis of the investigated subjects revealed some interesting findings. Two main clusters were identified. Cluster 1 comprised participants from all investigated groups (COPD, non-COPD smokers, and non-COPD non-smokers) and was characterized by lower sputum cell counts, lower inflammatory cytokine levels, and lower CHIT1, YKL-40, and chitinolytic activity in sputum. Cluster 2 was formed exclusively by patients with COPD and demonstrated more pronounced airway inflammation reflected by a higher sputum inflammatory cell count, higher cytokine concentrations, and higher CHIT1, YKL-40, and chitinase activity than Cluster 1. Interestingly, when only COPD patients from Cluster 1 and Cluster 2 were compared, both groups were comparable in terms of age, gender distribution, BMI, smoking history and pulmonary function, despite relevant differences in the sputum inflammatory profile as presented above. This supports the concept of treatable traits in obstructive lung diseases, highlighting the need to identify individual disease features in patients with similar clinical manifestations to tailor precise treatment. [24,25]. The extremely low values of CHIT1 and YKL-40 and reduced chitinolytic activity in COPD patients from Cluster 1 may be, at least partially, responsible for the lack of differences in chitinase levels/activity between the COPD group and controls.

One limitation of our study was the small COPD sample size used for cluster analysis. The COPD group was also too homogenous to reveal any significant differences in lung function and severity of the disease. Moreover, more prospective longitudinal studies should be conducted to characterize COPD phenotypes and assess if these clusters influence the risk of exacerbation and the effectiveness of inhaled therapy. Although in the current study, the level of airway inflammation was attributed to only one factor, i.e., tobacco smoke exposure, we could have missed other crucial factors possibly contributing to the level of airway inflammation (e.g., working in a toxic environment, passive smoking).

Another significant limitation is that the elderly participants in our study had different comorbidities which could have affected our results. CHIT1 and YKL-40 are not specific to lung pathology and their levels can be elevated in multiple conditions, such as cardiovascular disease, cancer, inflammation, diabetes, and others [3]. Chitinase expression and activity may also be influenced by genetic polymorphism, although literature data on this matter are not uniform.

## 5. Conclusions

In conclusion, elevated IS YKL-40 levels in the COPD group and the relationship between some IS chitinases/CLPs (YKL-40) and some proinflammatory cytokines (IL-8 and MMP-9) suggest that chitinases may play an active role in chronic inflammation and tissue remodeling. Our study points to an association between the expression of chitinases and CLPs in the airways and the level of airway inflammation due to cigarette smoke exposure. However, this relationship seems to be ambiguous and needs further elucidation. Two main clusters were identified in the study. Cluster 2 included only COPD patients and was characterized by higher chitinase levels, chitinolytic activity, and more intense airway inflammation than Cluster 1. This difference should be considered when planning personalized treatments and precision medicine strategies for chronic airway diseases.

## Figures and Tables

**Figure 1 cells-11-03765-f001:**
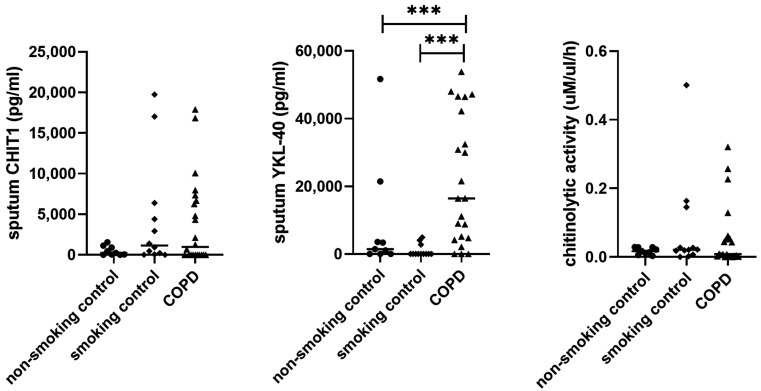
Induced sputum CHIT1 and YKL-40 concentrations and chitinolytic activity in COPD and control groups. Data are presented as median and individual values. Each data point represents a single individual categorized as follows: • represents cases from non-smoking control, ∎ represents cases from smoking control and ▲ represents COPD cases. *** *p* ≤ 0.001.

**Figure 2 cells-11-03765-f002:**
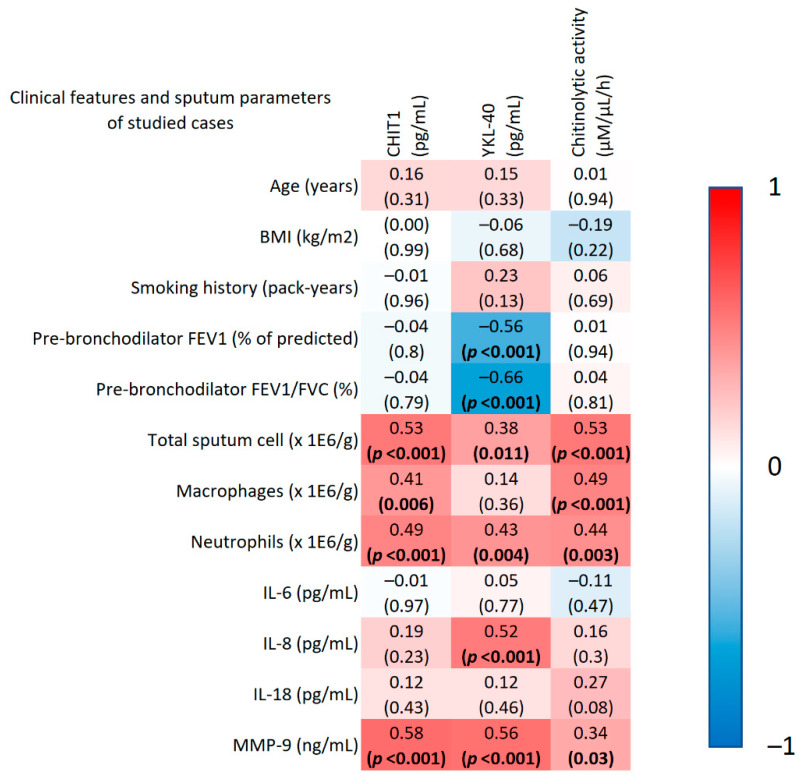
Correlation heatmap between selected clinical and sputum parameters and sputum chitinase levels in the entire study group (COPD patients and controls), expressed as correlation coefficient (r). The heatmap shows the correlation between clinical and sputum parameters (rows) and IS CHIT1 and YKL-40 levels and chitinolytic activity (columns). The colours of the heatmap correspond to the r-value (−1 to +1). The upper row in each square of the table denotes the r-value of the Spearman′s rank correlation and the lower row denotes its *p*-value. Abbreviations: BMI-body mass index; CHIT1-chitotriosidase; FEV1-forced expiratory volume in the first second; FVC-forced vital capacity; IL-interleukin; MMP-9-matrix metalloproteinase 9; YKL-40-chitinase-3-like protein 1. Bolded values show statistical significance (*p* < 0.05).

**Figure 3 cells-11-03765-f003:**
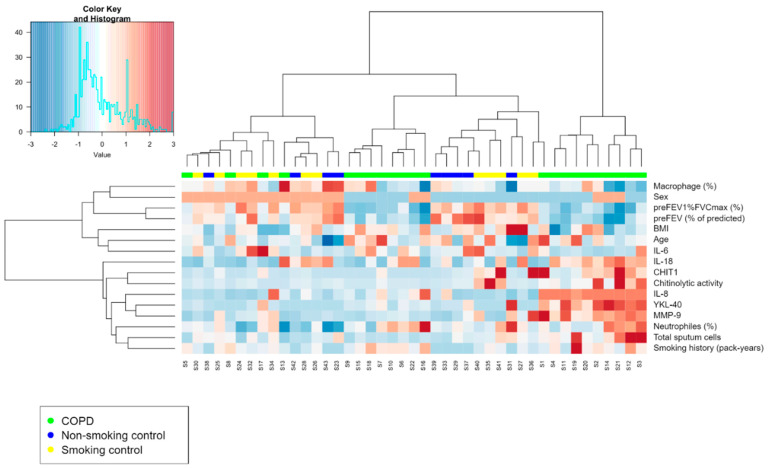
Hierarchical clustering heatmap of COPD patients and two control groups: non-smokers and smokers. The analysis included: clinical features, sputum cytokine concentrations, sputum chitinase levels and activity, and sputum cellular features. The vertical axis represents the study participants and the horizontal axis shows clinical and laboratory data. The colour scale represents the normalized value of a variable, with red representing the highest values.

**Table 1 cells-11-03765-t001:** Basic characteristics of the investigated participants.

Clinical Data	Non-Smoking Control (*n* = 9)	Smoking Control (*n* = 12)	COPD (*n* = 22)	*p*-Value
Sex (Male/Female)	4/5	7/5	13/9	0.61
Age (years)	53 (45–69)	62 (59.5–67)	63.5 (61–75)	0.1
BMI (kg/m^2^)	26.2 (24.7–30.1)	29 (26.5–32.4)	27.3 (23.5–31.1)	0.452
Smoking status (current smokers/ex-smokers)	n/a	9/3	12/10	0.47
Smoking history (pack-years)	0	30 (17.5–36.5)	45 (30–55)	**0.005 *^,&^**
Pre-bronchodilator FEV_1_ (% of predicted)	106 (83–115)	88 (83.5–94.5)	57.5 (49–62)	**<0.001 *^,&^**
Pre-bronchodilator FEV_1_/FVC (%)	74.62 (67.2–77.6)	72.98 (71.2–79.2)	47.4 (43.1–54.9)	**<0.001 *^,&^**
Post-bronchodilator FEV_1_ (% of predicted)	n/a	n/a	64 (55–73)	n/a
Post-bronchodilator FEV_1_/FVC (%)	n/a	n/a	51.38 (46–59.1)	n/a
GOLD A/B/C/D categories (n)	n/a	n/a	2/17/1/2	n/a
CAT score (points)	n/a	n/a	12 (8–17)	n/a
mMRC scale (points)	n/a	n/a	1 (1–2)	n/a

Abbreviations: BMI-body mass index; CAT-COPD Assessment Test; COPD-chronic obstructive pulmonary disease; FEV_1_-forced expiratory volume in first second; FVC-forced vital capacity; GOLD-Global Initiative for Chronic Obstructive Lung Disease; mMRC-modified Medical Research Council; n/a-not applicable. The significance for multiple comparisons is labeled as follows: * between COPD and smoking control, ^&^ between COPD and non-smoking control. Bolded values show statistical significance (*p* < 0.05).

**Table 2 cells-11-03765-t002:** Sputum parameters in study participants.

Parameters of Induced Sputum	Non-Smoking Control (*n* = 9)	Smoking Control (*n* = 12)	COPD (*n* = 22)	*p*-Value
Total cell count (×10^6^/g)	1.2 (0.7–2)	1.8 (1.4–2.2)	1.3 (0.7–2.5)	0.74
Macrophage cell count (×10^6^/g)	0.5 (0.3–0.8)	0.6 (0.5–0.8)	0.5 (0.21–0.9)	0.769
Macrophages (%)	46 (38–48)	39 (37–46.5)	35.5 (26–46)	0.395
Neutrophil cell count (×10^6^/g)	0.6 (0.3–1)	0.8 (0.6–1.2)	0.8 (0.4–1.2)	0.789
Neutrophils (%)	48 (40–50)	50.5 (40.5–58)	54.5 (45–68)	0.175
Lymphocyte cell count (×10^6^/g)	0.1 (0.1–0.1)	0.1 (0.1–0.2)	0.1 (0–0.1)	**0.017 ***
Lymphocytes (%)	5 (5–7)	7.5 (4–11)	3 (0–5)	0.107
Eosinophil cell count (×10^6^/g)	0.02 (0–0.04)	0.02 (0.02–0.05)	0 (0–0.02)	0.072
Eosinophils (%)	1 (1–2)	2 (1–4)	0.5 (0–1)	0.223
IL-6 (pg/mL)	1.8 (1.5–2.3)	3. (0.6–5.9)	0.8 (0.4–3.5)	0.217
IL-8 (pg/mL)	63.2 (44.1–111.7)	81.9 (51.5–95.4)	388.4 (149.8–500)	**0.01 *^,&^**
IL-18 (pg/mL)	27.1 (15–56.1)	14.2 (0–80.8)	60.5 (6.3–93.5)	0.361
MMP-9 (ng/mL)	66.8 (19.9–104.7)	144.4 (69.5–417.8)	258.4 (60.7–624.1)	0.175
CHIT1 (pg/mL)	238.5 (42.4–903.9)	1158.4 (135.8–5387.3)	961.8 (0–6669.1)	0.519
Chitinolytic activity (μM/μL/h)	0.017 (0.01–0.02)	0.021 (0.013–0.086)	0.01 (0.003–0.05)	0.74
YKL-40 (pg/mL)	1422.3 (0–3550.8)	0 (0–1378.7)	16,407 (4738.3–42,193.3)	**<0.001 *^,&^**

Abbreviations: CHITI1-chitotriosidase; COPD-chronic obstructive pulmonary disease; IL-interleukin; MMP-9-matrix metalloproteinase 9; YKL-40-chitinase-3-like protein 1. Values are expressed as median and IQR. The significance for multiple comparisons is labeled as follows: * between COPD and smoking control, ^&^ between COPD and non-smoking control. Bolded values show statistical significance (*p* < 0.05).

**Table 3 cells-11-03765-t003:** A logistic regression model including selected clinical and sputum parameters identifying risk factors for COPD.

Clinical and Sputum Paramters	r-Coefficient	*p*-Value
(Intercept)	1.02	0.11
IL-8 (pg/mL)	0.66	0.38
Smoking history (pack-years)	5.23	**0.01**
CHIT1 (pg/mL)	0.20	0.80
YKL-40 (pg/mL)	1.69	**0.02**
IL-6 (pg/mL)	−0.26	0.67
Chitinolytic activity (uM/uL/h)	−0.97	0.35

Results are expressed as correlation coefficient (r) and *p*-value. Abbreviations: CHITI1-chitotriosidase; IL-interleukinYKL-40-chitinase-3-like protein 1. Bolded values show statistical significance (*p* < 0.05).

**Table 4 cells-11-03765-t004:** Clinical characteristics of patients in two identified clusters.

Clinical parameters	Cluster 1 (*n* = 34)	Cluster 2 (*n* = 9)	*p*-Value
COPD, *n* (%)	13 (38)	9 (100)	**0.005**
Control smokers, *n* (%)	12 (35)	0 (0)	**<0.001**
Control non-smokers, *n* (%)	9 (27)	0 (0)	**<0.001**
Male/female	17/17	6/3	0.37
Age (years)	62.5 (56–75)	62 (61–72)	0.85
BMI (kg/m^2^)	27.9 (25.2–31.1)	24.8 (21.9–28.6)	0.17
Smoking history (pack-years)	26 (0–40)	45 (40–55)	**0.03**
Pre-bronchodilator FEV_1_ (l)	2.2 (1.7–3)	1.3 (1–1.8)	**0.008**
Pre-bronchodilator FEV_1_ (% of predicted)	89 (62–91)	50 (45–57)	**<0.001**
Pre-bronchodilator FEV_1_/FVC (%)	68.5 (55.8–74.6)	46.3 (42.5–50.8)	**<0.001**

Abbreviations: BMI-body mass index; COPD-chronic obstructive pulmonary disease; FEV_1_-forced expiratory volume in first second; FVC-forced vital capacity. Bolded values show statistical significance (*p* < 0.05).

**Table 5 cells-11-03765-t005:** Cellular characteristics and the levels of sputum mediators in the two main identified clusters.

Parameters of Induced Sputum	Cluster 1 (*n* = 34)	Cluster 2 (*n* = 9)	*p*-Value
Total cell count (×10^6^/g)	1.2 (0.65–2.03)	2.82 (2.11–5.9)	**<0.001**
Neutrophils (%)	50.5 (45–59)	54 (45–71)	0.263
Neutrophils (×10^6^/g)	0.64 (0.33–0.98)	2.14 (0.97–3.56)	**0.001**
Macrophages (%)	40 (35–48)	32 (22–37)	**0.04**
Macrophages (×10^6^/g)	0.49 (0.27–0.79)	0.85 (0.67–1.95)	**0.004**
IL-6 (pg/mL)	1.98 (0.72–4.96)	0.46 (0.19–0.79)	**0.02**
IL-8 (pg/mL)	82.8 (57.76–167.95)	500 (500–500)	**<0.001**
IL-18 (pg/mL)	18 (0–62.25)	87.34 (68.79–93.47)	**0.003**
MMP-9 (ng/mL)	76.89 (26.73–193.1)	624.12 (426.7–647.01)	**<0.001**
CHIT1 (pg/mL)	135.76 (0–1553)	6669.08 (2119.33–7963.19)	**0.002**
Chitinolytic activity (μM/μL/h)	0.013 (0.003–0.027)	0.06 (0.05–0.23)	**0.001**
YKL-40 (pg/mL)	1770.29 (0–5151.67)	46,429.17 (30,799.17–47,146.67)	**<0.001**

Abbreviations: CHIT1-chitotriosidase; IL-interleukin; MMP-9-matrix metalloproteinase 9; YKL-40-chitinase-3-like protein 1. Bolded values show statistical significance (*p* < 0.05).

**Table 6 cells-11-03765-t006:** Clinical characteristics of COPD patients in the two identified clusters.

Clinical Data	COPD from Cluster 1 (*n* = 13)	COPD from Cluster 2 (*n* = 9)	*p*-Value
COPD, *n* (%)	13 (38)	9 (100)	0.35
Male/female (*n*)	7/6	6/3	0.64
Age (years)	68 (62–77)	62 (61–72)	0.35
BMI (kg/m^2^)	27.9 (24.8–31.1)	24.8 (21.9–28.6)	0.3
Smoking history (pack-years)	45 (30–52.5)	45 (40–55)	0.815
Pre-bronchodilator FEV_1_ (l)	1.5 (1.17–1.9)	1.3 (1–1.8)	0.664
Pre-bronchodilator FEV_1_ (% of predicted)	61 (51–66)	50 (45–57)	0.082
Pre-bronchodilator FEV_1_/FVC (%)	47.5 (44.8–56.7)	46.3 (42.5–50.8)	0.256
Post-bronchodilator FEV_1_ (l)	1.7 (1.4–2.1)	1.4 (1.3–2)	0.815
Post-bronchodilator FEV_1_ (% of predicted)	68 (58–73)	57 (51–64)	0.285
Post-bronchodilator FEV_1_/FVC (%)	52.6 (47.5–59.1)	49.4 (44.4–51.2)	0.285

Abbreviations: BMI—body mass index; COPD-chronic obstructive pulmonary disease; FEV_1_-forced expiratory volume in first second; FVC-forced vital capacity. Bolded values show statistical significance (*p* < 0.05).

**Table 7 cells-11-03765-t007:** Sputum cellular characteristics and mediator levels in COPD patients from the two identified clusters.

Parameters of Induced Sputum	COPD from Cluster 1 (*n* = 13)	COPD from Cluster 2 (*n* = 9)	*p*-Value
Total cell count (×10^6/^g)	0.8 (0.5–1.1)	2.8 (2.1–5.9)	**<0.001**
Neutrophils (%)	55 (49–62)	54 (45–71)	0.69
Neutrophils (×10^6^/g)	0.46 (0.26–0.69)	2.1 (1–3.6)	**0.001**
Macrophages (%)	40 (32–47)	32 (22–37)	0.22
Macrophages (×10^6^/g)	0.2 (0.2–0.4)	0.9 (0.7–2)	**0.001**
IL-6 (pg/mL)	2.9 (0.6–5)	0.5 (0.2–0.8)	0.07
IL-8 (pg/mL)	150.1 (72.8–271.9)	500 (500–500)	**0.003**
IL-18 (pg/mL)	7.2 (3.7–53.4)	87.34 (68.8–93.5)	**0.019**
MMP-9 (ng/mL)	81.2 (54.2–149.8)	624.1 (426.7–647)	**0.001**
CHIT1 (pg/mL)	0 (0–0)	6669.1 (2119.3–7963.2)	**0.003**
Chitinolytic activity (μM/μL/h)	0.004 (0.001–0.01)	0.06 (0.05–0.23)	**0.001**
YKL-40 (pg/mL)	5151.7 (2118.3–11,130)	46,429.2 (30,799.2–47,146.7)	**0.001**

Abbreviations: CHIT1-chitotriosidase; COPD-chronic obstructive pulmonary disease; IL-interleukin; MMP-9-matrix metalloproteinase 9; YKL-40-chitinase-3-like protein 1. Bolded values show statistical significance (*p* < 0.05).

## Data Availability

The reported data are available from the corresponding author upon reasonable request.

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
