# Peer review of "The Role of Chitinases in Chronic Airway Inflammation Associated with Tobacco Smoke Exposure"

_cells, 2022, doi:10.3390/cells11233765_

Round 1
Reviewer 1 Report
The authors of the manuscript entitled “The role of chitinases in chronic airway inflammation associated with tobacco smoke exposure” have presented the role of CHIT1 and YKL-40 and their correlation with inflammatory cytokine levels in COPD patients. However, there are some concerns needed to be addressed in order to have a proper scientific understanding about the role of chitinases in COPD.
My comments are as below:
a) Authors should provide a rationale behind taking such a low number of COPD patients and then doing the cluster analysis.
b) It is already an ambiguous fact that the CHIT1 and YKL-40 plays a role of in inflammatory lung diseases including IPF, COPD, asthma etc., but it needs to be supported by some in vitro or in vivo studies to fully comprehend it. Thus, authors should include some data regarding these.
c) In addition to correlation studies, authors should include the logistic regression studies to fully elucidate the relation between co-founding factors.
d) The authors should state clearly mention (about Table 2), which groups where compared in order to calculate the p-value. Instead of table format, author may include bar charts to represent the statistical significance.
Reviewer 2 Report
A very interesting paper with some novel aspects of the chitinases role of pathogensis of COPD. I belive it should be accepted in the current form.
Author Response
Thank you very much for your positive opinion. We hope that the changes introduced to the text to comply with all the Reviewer’s’ opinions and suggestions allowed to further clarify and achieve better visualization of the results of our study.
Reviewer 3 Report
To the Editor,
In their interesting study, Przysucha N et al. analyzed the levels of CHIT1, chitinase-like protein (CLP) YKL-40, and chitinolytic activity (I will refer to these three as ‘chitinase markers’) in the induced sputum (IS) from patients with COPD, and smoking or non-smoking control subjects. They also compared these data to renown markers of airway inflammation in COPD. The study is potentially important because the mechanistic role of chitinases or their diagnostic usefulness in airway diseases are still poorly understood. Similarly, the precise functions of airway chitinases and CLPs are still unknown. In particular, it is not clear to what extent increased airway chitinase activity or CLPs are characteristics of COPD (disease trait; Letuve S et al. 2010) or more a response to noxious agents associated with smoking (eg, Majewski S et al. 2019; Seibold MA et al. [PMID:18845328]). The Authors did not show increased levels of CHIT1 or chitinase activity in IS from patients with COPD or smokers. On the contrary, the IS concentration of YKL-40 was markedly increased in COPD. Further cluster analysis revealed that higher levels of chitinase markers were restricted to a subset of patients with COPD (41%) who generally had more prominent inflammation in the airways, but showed similar clinical characteristics as those who grouped with controls. The Authors conclude that chitinases are involved in smoking-related airway inflammation and may be potentially used for COPD phenotyping (abstract). The association of chitinases with airway inflammation in COPD has already been described; however, data on how chitinases are related to disease clusters defined by the inflammatory pattern in the airways are novel.
Major
1. Data interpretation. Comparing clinical and airway variables in cluster-1 that is very heterogeneous (includes healthy smokers, non-smokers and COPD), to cluster-2 that includes only COPD, is somewhat inadequate (Tables 5 and 6). More relevant data comparing only COPD subphenotypes are already shown in Tables 7 and 8. Moreover, the Authors conclude (abstract) that analysis of chitinases may be a ‘potential novel marker in COPD phenotyping’. Chitinases appear to be only a part of inflammatory response associated with COPD and increase in cluster-2 as other innate cytokines. What is the diagnostic performance of chitinase markers (and others) in the detection of identified COPD subphenotypes? Are they unique or somehow better than other inflammatory biomarkers used?
Minor
1. Data presentation. I think the first part of the data could be easier to assimilate by the reader with the help of an additional figure showing the concentration and activity of chitinases in the studied groups (individual data points are required due to low sample size) and selected correlations (e.g., cross-correlation heat matrix). This will also help to highlight the issue of low-levels and heterogeneity (lines 308-309). Certainly, the data for the whole group are currently shown in the tables, so this is just a suggestion. However, it could be interesting to show whether the correlation trends still hold if the analysis is focused only on the COPD group.
2. Impact of covariables. I am very convinced by the model presented in which chitinases are overexpressed only in a subset of patients with COPD with overt airway inflammation. Furthermore, chitinase markers and other cytokines were found to be considerably intercorrelated (Table 4). Multivariate regression models could help explain the relationship between chitinase markers and important clinical and airway variables. This could clarify whether and to what extent some inflammation related (neutrophils) or clinical variables (eg, decrease in FEV1) explain the relationship between chitinases and airway inflammation. Again, for clarity, I suggest analyzing separately the dataset including only COPD patients.
3. Variables in the cluster analysis. I don’t quite understand the idea of including such different variables in this analysis, ranging from demographic (gender) to, say, IL-8. Because chitinases have already been described as airway biomarkers of airway inflammation, the Authors should consider including only airway/inflammation-related variables.
4. Treatment data are not provided. Steroids seem to have a considerable effect on chitinase expression. The patients were classified as GOLD B, so ICS use was probably not common. Were there differences in COPD scores (classes) and steroid treatment when comparing the two clusters?
5. One hypothesis was that airway levels of chitinase markers are not associated with ‘specific disease, i.e. COPD, but with the level of cigarette smoke exposure and the level of airway inflammation’ (65-67, 277-279). The authors validated ‘smoking’ part of this hypothesis by comparing healthy non-smokers with smokers, which did not reveal any differences in chitinase markers. Currently, the presented manuscript rather shows that increased chitinases are related to the level of inflammation in COPD (in fact it is a ‘specific disease’ trait) but not to smoke exposure itself. I am not convinced that this hypothesis was fully confirmed by data, as presented in the discussion section (282-283). This should be clarified anyway. I am curious if it is possible to somehow grade cigarette smoke exposure in the COPD group, that is, to stratify by the number of pack/years and the number of cigarettes smoked per day? Was there any difference in these parameters in comparison of COPD clusters?
6. In different human populations ~6% subjects are completely deficient for CHIT1 activity due to short duplication in exon 10, and around 35% of subjects are carriers with decreased activity (PMID:9748235). The deficiency is somehow compensated, as these subjects are generally healthy. Did the Authors check for CHIT1 gene variants, if so, was there any relationship between carrier status and chitinase activity? This problem has recently been observed in studies on chitinase in sarcoidosis (PMID:33307063). In the small study by Seibold MA et al. 2008 who compared smokers and non-smokers, BAL chitinase activity depended on the genotype, yet it tended to be higher in smokers also in the carrier group. Moreover, one should speculate that some disease subtypes (e.g., inflammatory clusters detected in this study) could be affected by CHIT1 genotype. The Authors should add a short comment if they believe this may have influenced the results?
7. The sentence ‘[…] cigarette smoke exposure measured as a number of inflammatory cells’ (325-326) is, of course, relevant for COPD pathogenesis and the data on how cigarette smoke affects airway cells. However, in this study cluster-2 was more inflammatory, yet it had similar smoke exposure as cluster-1. This suggests the contribution of other factors related to the pathogenesis of COPD inflammation, rather than smoking alone. Perhaps this conclusion should be more reserved.
Round 2
Reviewer 1 Report
none